# Automatic Roadway Features Detection with Oriented Object Detection

Hesham M. Eraqi [1,2,*], Karim Soliman [3], Dalia Said [3], Omar R. Elezaby [2], Mohamed N. Moustafa [1] and Hossam Abdelgawad [3]

1. Computer Science and Engineering Department, The American University in Cairo, New Cairo 11835, Egypt; m.moustafa@aucegypt.edu
2. Department of Driving & Comfort Assistance Systems, Valeo, Smart Village 12577, Egypt; omar.r.elezaby@gmail.com
3. Traffic and Highway Engineering Department, Cairo University, Cairo 12613, Egypt; eng.ksoliman@gmail.com (K.S.); daliasaid@eng.cu.edu.eg (D.S.); hossam.abdelgawad@alumni.utoronto.ca (H.A.)
* Correspondence: heraqi@aucegypt.edu

**Featured Application: (1) A pervasive system based on deep learning that can utilize a camera and GPS sensors of a car on a road to provide an up-to-date roadway safety estimation, which is a costly task in existing road asset management systems (RAMS). (2) A system that automatically estimates roadway safety to enable or disable vehicles' autonomous driving or advanced driving active safety (ADAS) functions.**

**Abstract:** Extensive research efforts have been devoted to identify and improve roadway features that impact safety. Maintaining roadway safety features relies on costly manual operations of regular road surveying and data analysis. This paper introduces an automatic roadway safety features detection approach, which harnesses the potential of artificial intelligence (AI) computer vision to make the process more efficient and less costly. Given a front-facing camera and a global positioning system (GPS) sensor, the proposed system automatically evaluates ten roadway safety features. The system is composed of an oriented (or rotated) object detection model, which solves an orientation encoding discontinuity problem to improve detection accuracy, and a rule-based roadway safety evaluation module. To train and validate the proposed model, a fully-annotated dataset for roadway safety features extraction was collected covering 473 km of roads. The proposed method baseline results are found encouraging when compared to the state-of-the-art models. Different oriented object detection strategies are presented and discussed, and the developed model resulted in improving the mean average precision (mAP) by 16.9% when compared with the literature. The roadway safety feature average prediction accuracy is 84.39% and ranges between 91.11% and 63.12%. The introduced model can pervasively enable/disable autonomous driving (AD) based on safety features of the road; and empower connected vehicles (CV) to send and receive estimated safety features, alerting drivers about black spots or relatively less-safe segments or roads.

**Keywords:** computer vision; object detection; road transportation; safety management

## 1. Introduction

Despite roadway safety and vehicle design improvements, the total number of fatal crashes still increases. The latest Global Status Report of the World Health Organization (WHO) reported an estimated 1.35 million annual deaths due to road traffic crashes worldwide [1]. Extensive research has been conducted to quantify the effect of roadway features on safety in the last two decades. Roadway features data collection and analysis is, not only an expensive and time-consuming process, but should also be carried out periodically to ensure properly maintaining roadway conditions. Therefore, automating roadway

features detection is a promising application to contribute to the body of roadway safety management efficiently and less costly, by harnessing the potential of modern artificial intelligence (AI) techniques.

In the last decade, a dramatic increase in computational power is witnessed, along with an abundant and growing amount of big data became available; the collective sum of the world's data volume is expected to grow from 33 Zettabytes this year to a 175 Zettabytes by 2025, for a compounded annual growth rate of 61% [2]. The field of AI greatly benefited from such two achievements to successfully train deeper, i.e., more complex, machine learning models that can learn efficient useful representations from data. That improvement is a by-product of learning feature maps [3] rather than hand-crafting them in traditional computer vision practices [4]. This led AI to exceed the human performance in some tasks like image classification, and to provide accurate object detection models from images [5]. Modern AI success can be a key enabler to the automation of roadway safety features detection.

The objective of this paper is to harness the potential of AI computer vision to automate the process of roadway safety features detection, as a step aiming to make the process more applicable and less costly. An oriented bounding box object detection model is introduced; it presents a modified version of a state-of-the-art deep learning-based object detection method. The model solves an orientation encoding discontinuity problem to improve deep learning-based oriented object detection. The model then feeds detections to a proposed rule- based roadway safety features detection module. The input to the proposed system is an image sequence generated from a vehicle-mounted front-facing camera and the GPS information, and the output is a list of ten essential safety features. Briefly, in a single drive, the proposed architecture attempts to automatically score roadway safety features without relying on expensive data collection, management, and analysis resources.

The automatic roadway safety features assessment can be used to pervasively enable or disable autonomous driving (AD) or advanced driver-assistance systems (ADAS). In case of connectivity, connected vehicles (CV) would send and receive estimated safety features and could send alerts to drivers about black spots or relatively less-safe segments or roads. Black spots identification is based on when the detected safety features scores are considerably lower than the roadway safety standards. Additionally, the proposed system could aspire for "safety-based navigation" applications where travelers can choose not only faster and shorter routes, but also safer routes. To validate the proposed approach, a case study dataset for roadway safety features extraction is introduced. The Greater Cairo Region (GCR) is the largest megacity in the Middle-East and North Africa (MENA) region [6]. The dataset covers 473 km travelled, between local roads, collector roads, regional and primary arterial roads, and regional and primary freeways. The dataset is composed of front-facing camera geotagged images fully annotated with object oriented bounding boxes and roadway safety features. The validation of the proposed approach is conducted on the test dataset, and promising results are achieved and reported as a baseline.

## 2. Background

### 2.1. Roadway Features in Relation to Safety

The cost of road crashes is huge, both economically and socially. Significant research around the world has focused on investigating the most significant features contributing to road safety. In this subsection, different roadway features are reviewed to quantify their contribution to road safety. Factors affecting crashes differ based on the driving environments. For example, in North America research efforts focused on geometric features and roadway elements, and found that horizontal alignment, lane, shoulder and traffic volumes significantly affect road safety; while in Africa features such as signs, marking, barriers, shoulder and right of way significantly affect road safety. In Asia, horizontal alignment and right of way were the most significant features affecting road safety. In Europe and New Zealand, horizontal alignment, vertical profile and lanes were

the most significant features. The most notable roadway features that affect safety on roads can be summarized as follows (categorized by geography):

- Canada [7]: traffic volume, lane width, access-point density, and segment length.
- US [8–17]: number of horizontal curves, curves radius, lane width, shoulder width, tangent length, annual average daily traffic (AADT), speed limit, median presence, hazardous elements on the roadside, vertical curvature, and road classification.
- Italy [18]: segment length, AADT, lane width, curvature indicator, and vertical grade.
- In England and Wales [19] and in New Zealand [20]: curvilinear alignments and road curvature.
- India [21,22]: roadside clearance, lighting, shoulder maintenance, and signs.
- Malaysia [23]: horizontal curvature, shoulder width, roadside activity, and median presence.
- China [24,25]: presence of heavy vehicles and non-motor vehicles, road density, trip frequencies, and land use types.
- South Korea [26]: presence of sidewalks and lighting and sufficient sight distance.
- Egypt [27–30]: shoulder width and type, roadside activities, annual average daily traffic (AADT), presence of motorcycles or heavy vehicles, signs, marking, lane width, median, barrier, lighting, and pavement condition.
- In Nigeria [31,32]: signs, marking, roadside fixed objects, shoulders, and pavement condition.

### 2.2. Oriented Object Detection

Object detection algorithms locate objects in camera images, classify the type of each object, and determine its location. Semantic segmentation algorithms classify each pixel of a camera image, or several pixels, into a class. The two methods have witnessed considerable improvements with the adoption of deep learning-based techniques [33] and achieved reliable results in many of real-world vision tasks, making them excellent candidates for automatic roadway safety features extraction. This subsection provides a review on relevant computer vision algorithms and discuss their applicability to the problem at hand.

Compared to semantic segmentation, object detection methods are more applicable for this research for three reasons: (1) the higher complexity of the semantic segmentation and the increased difficulty to automatically process and analyze its outcomes; (2) the high data annotation effort to train semantic segmentation methods; and (3) object detection methods can operate in real-time [5] efficiently when compared to semantic segmentation. Table 1 summarizes the relevant literature concerning deep learning-based object detection against three specific criteria to the problem of roadway feature detection: real-time, oriented, and undirected oriented.

Most modern object detectors that are not required to work in real-time rely on a two-stages detection approach having a region proposal stage and a detection stage as in R-CNN (Regions with Convolutional Neural Network features) [34], Fast R-CNN [35], and Faster R-CNN [36]. The first stage proposes a large set of regions that aim to include objects' hypotheses in the image; while the second stage aims to select the regions that actually contain objects out of the proposed hypotheses, and to classify these objects. In one-stage detectors, the region proposal stage and the classification and bounding boxes fine-tuning stage are conducted with a single network, which makes this approach much faster and more convenient for real-time applications. YOLO, or YOLOv1, [37] and SSD (Single Shot Detector; [38]) are the first methods to introduce such one-stage approach. YOLO9000 or YOLOv2 [39] is faster and more accurate than YOLOv1. YOLOv3 [5] is another obvious example of one-stage detectors that produce accuracies that are close to the two-stage detectors, while having the advantage of operating in real-time. In summary, the majority of real-time applications, as in our case, rely on one-stage detectors [40] as they are faster than two-stage detectors [41].

A key challenge to this research is the fact that the above-mentioned methods detect still, or horizontal, bounding boxes for objects in an image. However, the research in hand

requires the detection of oriented, or rotated, bounding boxes. In this study, there are classes with a large difference between the width and the height while appearing rotated in an image captured by a vehicle front-facing camera, such as road lane markings, side barriers, median, curbs, and billboards. Learning such oriented object detection task with a traditional object detector is a challenge. On one hand, the detector will suffer from limited accuracy; it is challenging for the network to learn the object nature since the object ground-truth horizontal box is not tightly fitting the object and contains a considerable amount of background information. On the other hand, during inference, it will be hard to determine the object shape and location from the predicted loose horizontal bounding boxes.

Few attempts that require oriented object detection were investigated in the literature. The common method is to extend one of the ordinary detectors discussed earlier by adding an additional orientation angle variable to be predicted. This method allows the model to predict the orientation angle of the detected bounding boxes in addition to their dimensions. An example for such method is in [42], where text is detected in images. Another example is in [43], where oriented objects in satellite images are detected. The method in [43] uses multi-angle prior anchor boxes to improve prediction accuracy. Such direct encoding of orientation angle leads to singularities that limit the model accuracy, which are avoided in [44] by predicting the sin and cos components of the angle encoded as a complex number (to detect objects from birds-eye-view Laser Scanner maps). The aforementioned three methods predict directed oriented objects, while this study tackles undirected oriented objects. For example, in the undirected case used in this study, it is equivalent to detect a lane marking in a front-facing camera image with an orientation of $45°$ or $225°$. However, in the directed object detectors case, detecting a car in a top-view image or map with orientation of $45°$ is not equivalent to a $225°$ of orientation. In [45], the authors tackle the problem of orientation encoding discontinuous by transforming angular prediction from a regression problem to a classification task, but the drawback is that the predicted orientation is provided in coarse-granularity. Other two-stage detectors as in [46] and [47] offer oriented object detections, but they are slower than one-stage detectors.

In [48], YOLOv1 is extended to detect undirected oriented objects by predicting quadrilaterals to detect text from images, and in [49], it is extended to detect size-independent polygons with a varying number of vertices defined on a polar grid. Nevertheless, these two methods make the detection problem harder for the model to learn in case of detecting only rectangle-shaped objects, because the model prediction search space include considerably large invalid options for non-rectangular objects. In addition, the method in [48] was found to have common failure cases with vertical text objects in images. Additionally, the aforementioned five oriented object detection methods do not adopt YOLOv3 concept of multiple detection at multiple scales, which promises better detection of small objects. In this paper, the relevant detectors are investigated, and a new method is introduced to address the application in hand—as detailed in Section 3.2.

**Table 1.** Object detection literature.

| Method | Year | Real-Time | Oriented | Undirected Oriented | Remarks on Method, Accuracy, and Speed |
|---|---|---|---|---|---|
| R-CNN [34] | 2014 | | | | First stage adopts Selective Search (Uijlings et al., 2013). |
| Fast R-CNN [35] | 2015 | | | | Faster than R-CNN due to sharing CNN computations between proposed regions. |
| FasterR-CNN [36] | 2015 | | | | Faster than Fast R-CNN by using Region Proposal Network (RPN) |
| YOLOv1 [37] | 2016 | √ | | | The first to use a single deep neural network and works in real-time at 45 FPS |
| SSD [38] | 2016 | √ | | | More accurate than YOLO, does not split the image into grids, instead it predicts offsets of anchor boxes for every location in a pyramidal feature maps hierarchy. Its SSD300 variant is faster than YOLO. |

**Table 1.** *Cont.*

| Method | Year | Real-Time | Oriented | Undirected Oriented | Remarks on Method, Accuracy, and Speed |
|---|---|:---:|:---:|:---:|---|
| YOLOv2, [39] | 2017 | ✓ | | | Faster and more accurate than YOLO by using anchor boxes. Its YOLOv2 416 × 416 variant is faster and more accurate than SSD300 |
| RetinaNet [50] | 2017 | ✓ | | | Matches the speed of the previous real-time methods while being more accurate than previous methods by using a focal loss function |
| YOLOv3 [5] | 2018 | ✓ | | | On par with RetinaNet accuracy but around four times faster and more accurate, faster than YOLOv2 by incorporating multiple improvements including multiple detection at multiple scales |
| [42] | 2017 | ✓ | ✓ | | Extends YOLOv1 by direct orientation angle regression to detect text from images |
| [43] | 2017 | ✓ | ✓ | | Similar to YOLOv1 with adding prior anchor boxes to detect objects in satellite images |
| [44] | 2018 | ✓ | ✓ | | Extends YOLOv2 by orientation angle regression represented as a complex number to detect objects from birds-eye-view Laser Scanner maps |
| [45] | 2020 | ✓ | ✓ | | Solves the problem of orientation encoding discontinuous by transforming angular prediction from a regression problem to a classification task but provides coarse-granularity in orientation prediction. |
| [48,49] | 2018, 2020 | ✓ | ✓ | ✓ | Reference [48] extend YOLOv1 to predict quadrilaterals encoding orientation to detect text from images, and [49] extends it to predict polygons with a varying number of vertices. Both methods are not optimal for rectangle-shaped objects. |
| Proposed Method | | ✓ | ✓ | ✓ | Extends YOLOv3 to predict undirected rectangle-shaped objects |

## 3. System Overview

Figure 1 introduces the proposed system framework. The development phase is composed of oriented object detector training and validation, and roadway safety rules validation. In the oriented object detector training and validation phase, the collected dataset is annotated with oriented bounding boxes of over 17 different object classes and ten safety features, the selection of these objects and features is determined based on the road safety comparative review summarized in Section 2.1. The dataset (detailed in Section 4.2) is split into training, validation, and testing subsets. The training data is passed through a data augmentation phase and the ground-truth labels are encoded to train a variant of YOLOv3 [5] object detector adopted to this study. The evaluation results of the validation data are used to fine-tune the learning process hyperparameters, and along with the evaluation results of the training data, the deep neural network object detector iterative learning process stopping criterion is examined.

The automatic roadway safety features assessment can be used to pervasively enable or disable autonomous driving (AD) or advanced driver-assistance systems (ADAS). The system can work reactively in case of no connectivity by relying on the self-vehicle camera and GPS. In case of connectivity, connected vehicles would send and receive estimated safety features targeting ubiquitous transposition network coverage and achieve higher accuracy by confirming and reinforcing detected features from multiple vehicles. Fusing safety features estimations for a roadway sector based on data coming from multiple vehicles, like averaging the scores, can make them more accurate, and allow for sending alerts to drivers about black spots or relatively less safe segments or roads. With modern prosperity in connected car technologies [51], sensor data coming from vehicles roaming the geographical area under deployment and the estimated safety features are aggregated in a remote server or in one host vehicle from the vehicles in the area for fusion and distribution.

To reduce information security risks, a validation entity such as a motorway infrastructure operator can receive and validate the vehicles' shared information. Additionally, the proposed system could aspire for "safety-based navigation" applications where travelers can choose not only faster and shorter routes, but also safer routes.

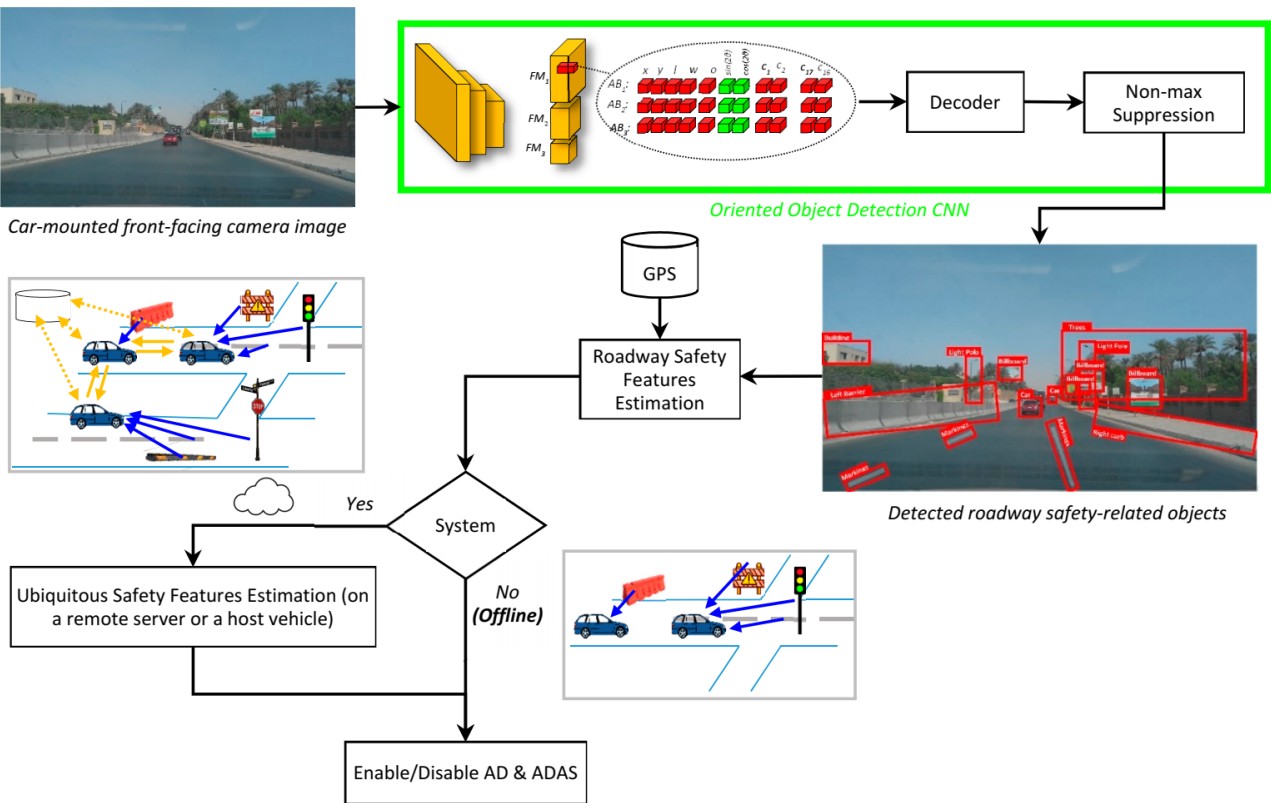

**Figure 1.** System overview during deployment phase.

Among the most important objects in road images in the context of safety analysis are road curbs, barriers, and lane markings. These objects appear oriented when seen from a front-facing view from a vehicle driver's perspective, and orientation is undirected ($\theta \equiv \theta + 180°$, where $\theta$ is the orientation angle). The majority of existing object detection methods from images, as detailed in the survey in Section 2.2, presume that detected objects appear horizontal in images. Adopting such methods while having oriented objects as in our case, produces objects bounding boxes predictions that are too loose, i.e., the bounding box is not representing the object shape nor position, big parts of the detected area cover other objects. Such downside effect is demonstrated in the results in Section 5. To overcome such limitations, a new model is developed to detect undirected oriented objects as detailed next.

### 3.1. Model Architecture

As one of the best one-stage object detectors in terms of detection accuracy and speed, YOLOv3 model architecture [5] was adopted as the base detector in this research. YOLOv3 divides the image into a grid of S × S cells, and for each grid cell it produces nine different predictions of different scales and sizes for the potential object centered at the grid cell. Each prediction includes a score that represents the confidence of the object existence in the corresponding cell. A list of anchor boxes defining the most common object bounding boxes' scales is pre-defined, and the network predicts the scale offsets from such anchor boxes. The base architecture of YOLOv3 was modified to predict undirected oriented object bounding boxes instead of horizontal bounding boxes.

Transfer learning [52] is applied. YOLOv3 was initially trained on MS COCO (Common Objects in Context) dataset [53], which is a rich object detection dataset featuring 123,287 training images representing 80 classes, and hence allows YOLOv3 CNN pipeline to detect a generic set of image objects features with a high precision. Then, the trained neural network was transferred from that broad domain to the specific one focusing on scenes from the roadway dataset. When training on the dataset, instead of starting the model iterative learning process from a random initial point, only the prediction layer is set to start from a random initial point, while the CNN pipeline starts from the knowledge gained from training on MS COCO dataset. The transfer learning is found necessary for the introduced model to converge to a reasonable detection performance; especially that the fine-tuning dataset is smaller than MS COCO dataset.

### 3.2. Orientation Representation

There is a variety of approaches to encode the orientation angle $\theta$ to the neural network as a ground-truth to learn. The straight-forward approach is to directly predict it as a single variable $\theta$ having a range of $[0, \theta_{max})$. Another approach is to predict two values encoding $\theta$; $\sin(\theta)$ and $\cos(\theta)$. These methods are not applicable to the research question herein due to an orientation encoding discontinuity problem that is characterized by a sudden change in the value encoding orientation ground-truth between objects having a slight orientation change. To better illustrate the orientation encoding discontinuity problem, Figure 2 shows for each encoding strategy several objects having different orientations covering the full $\theta$ range in $[0, 180°)$. The object rectangle color represents the orientation encoding value, blue for the smallest and red for the largest value.

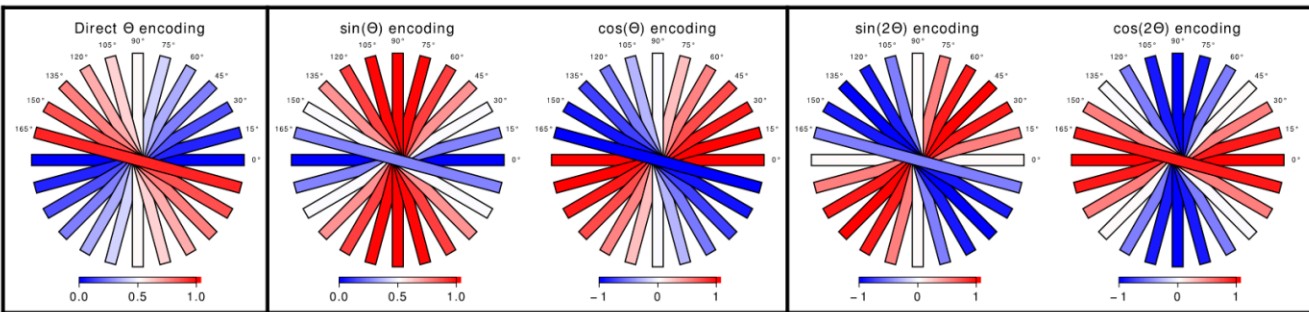

**Figure 2.** Undirected oriented objects orientation encoding strategies. The object rectangle color represents the orientation encoding value.

Such encoded value is the ground-truth that the neural network uses to learn orientation representation. This subsection explains the differences between these three encoding strategies and their limitation in application in roadway safety feature detection.

1. Direct orientation angle encoding: The $\theta$ range is normalized to the range of $[0, 1)$. Each oriented bounding box prediction has five descriptors, center point x-value and y-value, box length and width, and orientation angle $\theta$. Instead of width and height descriptors adopted in YOLOv3, the length (longest dimension) and width descriptors are adopted, which is a suitable choice to describe oriented bounding boxes. The length and width object descriptors are normalized by the image diagonal length. The region to the left in Figure 2 demonstrates the direct orientation angle encoding. In such case, orientation encoding discontinuity problem appears in the sudden change in the encoded value between the objects with orientations $0°$ and $179°$. Despite that these two objects have a very similar orientation; the network receives a drastically different orientation ground-truth. Neighboring orientations encoded values should be similar.

2. Orientation encoding with $\sin(\theta)$ and $\cos(\theta)$: Angles representation is periodic by nature. Towards the beginning and the end of the angle domain the prediction ground-truth should have a consistent value. To allow for this, the option of encoding

the orientation angle θ with sin(θ) and cos(θ) is investigated. This makes the problem easier for the network to learn, instead of predicting orientation directly, a predictor is learning object perpendicularity (sin(θ) predictor) and another one is inclination (cos(θ) predictor). Figure 2, middle region, demonstrates this encoding method. The problem of orientation encoding discontinuity is not happening for the sin(θ) predictor. However, the problem appears in the case of cos(θ) predictor, where there is a sudden change in encoding between the two objects with orientations of 0° and 179° (cos(0°) = 1, cos(179°) ≈ −1), despite their very similar orientation. Relying only on the sin(θ) predictor will introduce ambiguity to the network because it is symmetric around the orientation angle of 90°.

3. Orientation encoding with sin(2θ) and cos(2θ): To solve the discontinuity problem, encoding the orientation angle θ by the sin and cos components of double the orientation angle 2θ is proposed. This expands the domain of the prediction from [0, 180°) to [0, 360°) and the orientation encoding discontinuity problem is avoided as demonstrated in the right region of Figure 2. Like the case of encoding with sin(θ) and cos(θ), six descriptors are adopted for each oriented bounding box. Both of the sin(2θ) and cos(2θ) predictors are required, because sin(2θ) is symmetric around 45° and 135° of orientation, while cos(2θ) is symmetric around 90°.

Therefore, a solution using orientation encoding with sin(2θ) and cos(2θ) is proposed to better represent the road objects. The proposed model adopts YOLOv3 residual skip connections and up-sampling [5] to provide predictions on three different scales, which balances the model efficiency between detecting small and big objects. The grid cells in each scale are $52 \times 52$, $26 \times 26$, and $13 \times 13$ respectively. The network output layer tensor shape for each scale is $S \times S \times 3 \times (4 + 2 + 1 + 17)$, where $S \times S$ is the number of grid cells in that scale, each cell has 3 anchor boxes and each box has 4 offsets descriptors, two descriptors for sin (2θ) and cos (2θ), a confidence score, and the 17 classes in our dataset. In Figure 1, $FM_n$ is the nth output layer (feature map) of the CNN and $AB_n$ is the nth anchor box. x, y, l, w, and o are the predicted object *x*-axis offset, *y*-axis offset, length, width, and prediction confidence respectively. sin(2θ) and cos(2θ) represent the orientation encoding, where θ is predicted object orientation angle. $c_n$ is the probability that the predicted object is classified as the nth roadway safety-related object class. The decoder translates the $FM_1$, $FM_2$, and $FM_3$ outputs into a list of undirected oriented bounding-boxes, each box is for a detected roadway safety-related object.

### 3.3. Data Augmentation

During model training, online data augmentation was applied on half of the mini-batch images before feeding them to the network. To augment an image, it is passed through a pipeline of a sequential series of augmentation methods. Each augmentation method in the pipeline has a predefined probability of occurrence and has stochastic parameters to augment images differently. Two augmentation methods are adopted as shown in Figure 3; the first type is for photometric transformations (e.g., changing brightness, lighting conditions, and applying additive white Gaussian noise); and the second is for geometric transformations (e.g., horizontal flipping and rotation). In the case of horizontal flipping, object annotations of some classes should be flipped as well, like changing left curbs to be right curbs. Figure 3 illustrates sample outputs from the augmentation pipeline.

### 3.4. Model Training

The model training is conducted using a backpropagation learning algorithm [54], and Adam (short for adaptive moment estimation) optimizer [10] is used to optimize the model. Cross-validation is applied to ensure a reasonable model training stopping criterion. To evaluate the model performance, the mean average precision (mAP) is calculated for each epoch. For mAP calculation, Related IoU (ArIoU) estimation [43] is used instead of the exact IoU calculation, which is considerably slower in the case of oriented objects. In addition, ArIoU is found to be a good estimate of the exact IoU. However, unlike in [43], during

inference time the exact IoU calculation is used to produce more accurate object detections through the non-maximum suppression stage. The validation data mAP progression over training epochs in addition to the training data loss curve are used to define the training process stopping criterion. Once the validation data mAP trend decreases while the model is improving training data loss, training is stopped to avoid over overfitting. Binary cross-entropy loss is used for the class predictions, in addition to sum of squared error loss for all the other predicted descriptors.

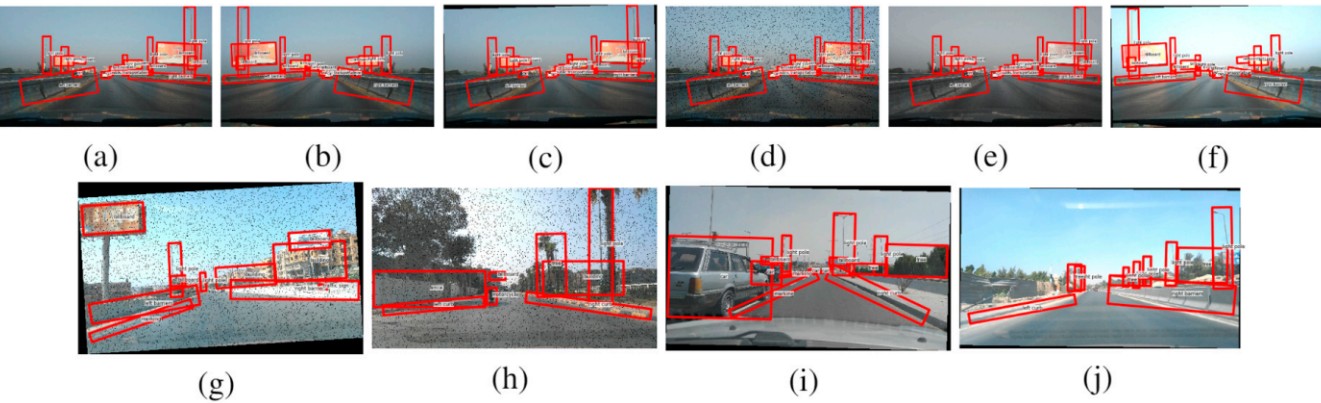

**Figure 3.** Samples of data augmentation. (**a**) Shows original image sample, and from (**b**–**e**) the effect of each augmentation method is shown; (**b**) horizontal flipping, (**c**) rotation, (**d**) adding noise, (**e**) changing brightness, and (**f**) shows the final augmentation pipeline result. (**g**–**j**) show four additional different output samples from the augmentation pipeline.

### 3.5. Roadway Features Extraction

As presented in Section 2.1, there are various features affecting roadway safety. To automate the process of roadway safety features detection, a rule-based module that automatically scores safety features is adopted based on literature in addition to the local environment that affects roadway safety. Figure 4 lists the roadway safety-related objects that are considered as input to the module. Due to lack of roadway safety data in developing countries, previous research attempted developing safety indicators for decision makers such as traffic safety equity [55]. Ten features were the criteria of the roadway safety rule-based module that scores and evaluate roadway safety features, where each feature was considered to have either a high, a medium, or a low safety score based on its impact on road safety expressed as a key safety indicator (KSI). As an example, the 'traffic condition' feature can be either fluid, dense, or congested based on speed and traffic density. A congested traffic condition leads to a low safety score and heavy traffic leads to more crashes on weekdays [56]. Moreover, the 'area type' feature can be either be rural, semi-urban, or urban based on buildings density. A rural area leads to a low safety score. 'Median' feature describes whether the road is divided or undivided based on the existence of a left barrier and left curb. 'Roadside barrier' feature is present based on the existence of a fence or right barrier objects. 'Existing marking', 'traffic signs', 'roadside barriers' and 'light poles' features have a positive safety impact, and therefore a higher "score". Moreover, 'existing billboards', 'pedestrian waiting area', and 'heavy vehicle' feature leads have a negative safety impact and therefore are given a lower "score". The weighted average of safety scores over the different features can be calculated for a road segment resulting in an estimate to the road segment safety. To summarize, we use a rule-based method of weighted averaging the safety scores over the different road features in a road segment to estimate its safety score. The score is a contentious number normalized to be from 0 to 1 (for a very safe segment).

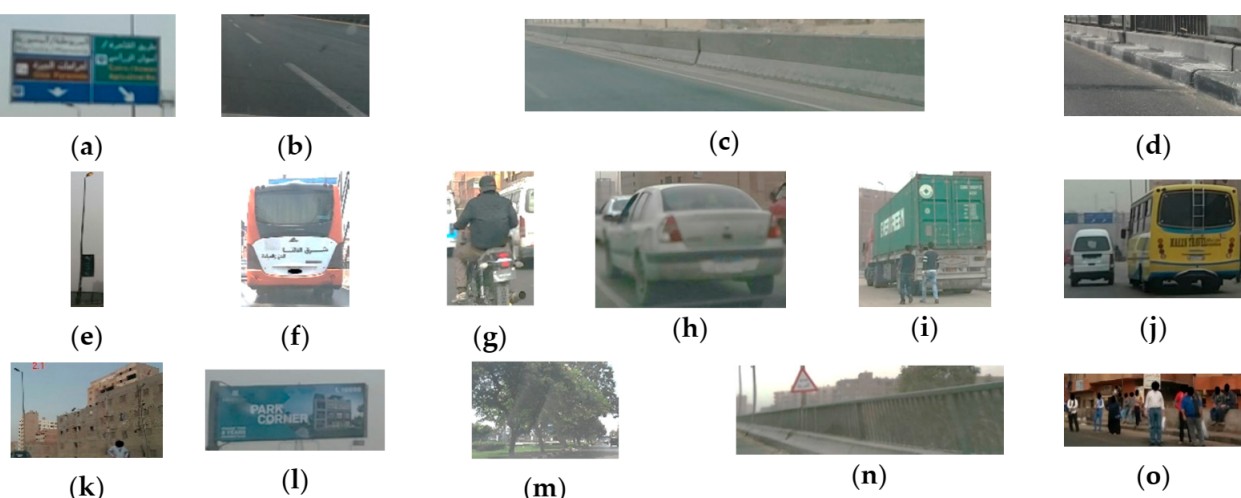

**Figure 4.** Roadway objects, from left to right and top to bottom: (**a**) traffic sign, (**b**) marking, (**c**) left and right barriers, (**d**) left and right curbs, (**e**) light pole, (**f**) bus, (**g**) motorcycle, (**h**) car, (**i**) heavy vehicle, (**j**) public transportation, (**k**) buildings, (**l**) billboards, (**m**) trees, (**n**) fence, and (**o**) pedestrian waiting areas.

## 4. Case Study

### 4.1. Greater Cairo

GCR is the largest megacity in the MENA region with a population of 24.5 million (according to Central Agency for Public Mobilization and Statistics—CAPMAS), hosting 3.7 million registered vehicles (in 2017) resulting in almost 2.5 million daily trips (6). Statistics show that 3604 crashes occurred on GCR roads in 2016, among them 1390 fatalities occurred. An equally startling statistic is that there are 4 deaths in Egypt per 100 km roads [57] while the rates of death in the UK and US are 0.47 and 0.92 people, respectively [1].

Despite the dire need for accurate crash data to methodically advance roadway safety research in Cairo/Egypt (and most developing countries), crash data, such as exact location, crash detailed report, and black spots, are largely missing. The proposed system in this paper can enable characterizing roadways by assigning a safety score which could support in screening roadway networks; such screening can further identify segments of low safety scores that either require further safety studies by decision makers or notify road users ahead of time when approaching such segments.

### 4.2. Dataset Design

The dataset was collected in the GCR and is composed of 473 km travelled between local roads, collector roads, regional and primary arterial roads, and regional and primary highways, as shown in Figure 5. The dataset was fully annotated with oriented object bounding boxes. Table 2 details the distances covered for each roadway class, such variety of road classes in the dataset allowed validating the proposed methods on roadway classes that are unseen during training and development time. A front-facing camera was designed to capture an image every 100 m, with each image including the longitude, latitude, altitude, time and average vehicle speed. The data collection plan was designed to cover a period of over two years from January 2017 to August 2019 and covered times from 6:00 a.m. to 12:00 p.m. and from 2:00 p.m. to 7:00 p.m. The complete dataset is composed of 4732 samples (divided to 3714 for training and validation and 1018 samples for testing) with a resolution of 2560 × 1440 pixels captured every 100 m. In order to label the collected images, a multi-platform object oriented bounding boxes (OBB) annotation tool was developed.

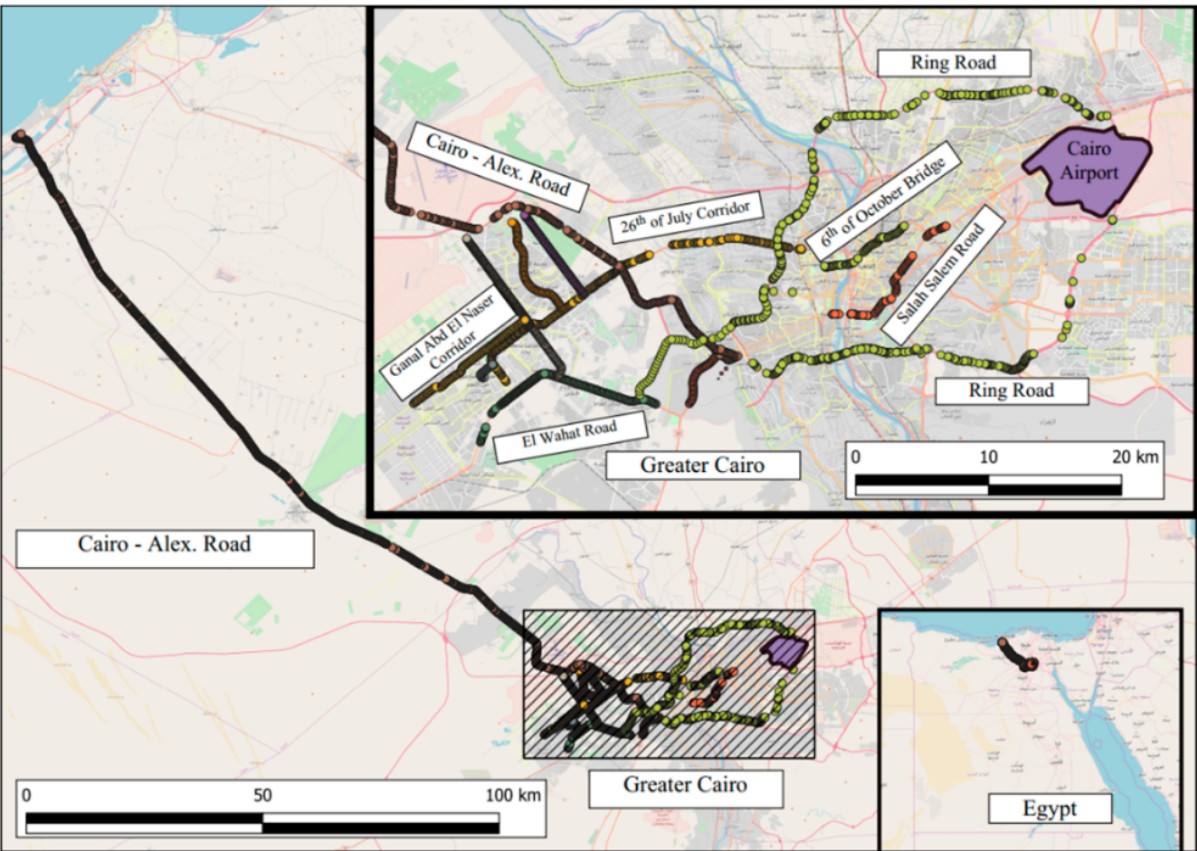

**Figure 5.** Data collection design in the Greater Cairo Region (GCR).

**Table 2.** Distances covered and number of snapshots for each road functional classification.

| Roadway Functional Classification | Kilometers Travelled | Number of Snapshots | Percentage in the Dataset |
|---|---|---|---|
| Primary Highways | 176 | 1760 | 37.19% |
| Regional Primary Highways | 164.1 | 1641 | 34.68% |
| Primary Arterial Roads | 74.2 | 742 | 15.68% |
| Secondary Arterial Roads | 21.5 | 215 | 4.54% |
| Primary Collector Roads | 29 | 290 | 6.13% |
| Local Roads | 8.4 | 84 | 1.78% |
| Total | 473.2 | 4732 | 100% |

## 5. Analysis and Results

Experiments are conducted on the testing dataset and the mean average precision (mAP) is reported based on 50% Intersection over Union (IoU). On an NVIDIA Titan X GPU, the proposed system processes camera images at 36 FPS. The results are presented in this section across two main verticals: oriented objects detection performance and accuracy and roadway safety features detection and score.

### 5.1. Oriented Objects Detection

Table 3 reports mAP-50 percentage results of the four different models for object detection detailed in Section 3.2:

- Model 1 is the original YOLOv3 [5] model trained on our dataset without the angle orientation.
- Model 2 adopted the direct orientation encoding.
- Model 3 encodes the orientation angle θ as sin(θ) and cos(θ).

- Model 4 is the developed model adopting sin(2θ) and cos(2θ) encoding of the orientation angle; while not using data augmentation during training.
- Model 5 is the developed model adopting sin(2θ) and cos(2θ) encoding of the orientation angle.

**Table 3.** Map score for each experiment.

| Model Number | Model | mAP [%] |
|:---:|:---:|:---:|
| 1 | No orientation [5] | 33.4 |
| 2 | θ | 50.0 |
| 3 | sin(θ) and cos(θ) | 50.7 |
| 4 | sin(2θ) and cos(2θ) without data augmentation | 43.1 |
| 5 | sin(2θ) and cos(2θ) | 50.3 |

Data augmentation is found to increase the mAP; model 5 achieves 7.2% higher mAP than model 4. Comparing the best performing model (model 5) with the original YOLOv3 model (model 1) shows an improvement of 16.9% mAP in detecting undirected oriented objects. On GCR test dataset, the introduced model achieves a 50.3% mAP. In our view, this is a significant achievement as the YOLOv3 is trained on MS COCO Common Objects in Context dataset which is more than 40 times larger than the current research dataset and achieves a 57.9% mAP, not to mention the added complexity and challenges of detecting oriented objects compared to the horizontal objects in COCO dataset.

As reported in Table 3, the 'sin(θ) and cos(θ)' model achieves the best overall mAP result. However, by inspecting the individual classes average precision (AP) as shown in Table 4, the 'sin(2θ) and cos(2θ)' model (model 5) is demonstrated to perform best for all the oriented classes like lane markings and curbs. Objectively, model 5 achieves more accurate orientation predictions than model 3, as shown in the random test data samples results in Table 5, as in sample numbers 3, 5, 6, 8, and 9. Accurate orientation prediction reduces the amount of redundant boxes corresponding to the same actual object, as duplicates have slight θ differences, and hence produce large IoUs that makes them eliminated as duplicates during non-maximum suppression. This demonstrates the effectiveness of the introduced 'sin(2θ) and cos(2θ)' model. On the other hand, the other non-oriented classes predictions did not benefit from such method. Table 4 also reports the object class distribution in the dataset. Marking, light pole, building, billboard, car, tree, and public transportation classes are highly represented, while traffic sign, right barrier, left barrier, right curb, left curb, pedestrian, and heavy vehicle classes appear less, and fence, bus, and motorcycle are the least represented classes in the dataset.

**Table 4.** Average precision per class for different models.

| Class | Appetences | Oriented Class | Average Precision [%] | | | | |
|:---:|:---:|:---:|:---:|:---:|:---:|:---:|:---:|
| | | | Model 1 (No Orientation) [5] | Model 2 (θ) | Model 3 (sin(θ) and cos(θ)) | Model 4 (sin(2θ) and cos(2θ) without Augmentation) | Model 5 (sin(2θ) and cos(2θ)) |
| | | | Oriented Classes | | | | |
| Marking | 6494 | Yes | 00.01 | 40.94 | 62.45 | 55.30 | 62.92 |
| Right Barriers | 1827 | Yes | 22.53 | 43.38 | 55.84 | 54.25 | 56.31 |
| Left Barriers | 1946 | Yes | 39.14 | 93.9 | 91.30 | 88.68 | 91.96 |
| Right Curb | 1091 | Yes | 1.45 | 36.73 | 43.18 | 39.23 | 51.72 |
| Left curb | 1215 | Yes | 0.54 | 64.74 | 71.13 | 59.05 | 73.83 |

**Table 4.** *Cont.*

| Class | Appetences | Oriented Class | Average Precision [%] | | | | |
|---|---|---|---|---|---|---|---|
| | | | Model 1 (No Orientation) [5] | Model 2 (θ) | Model 3 (sin(θ) and cos(θ)) | Model 4 (sin(2θ) and cos(2θ) without Augmentation) | Model 5 (sin(2θ) and cos(2θ)) |
| | | | Non-Oriented Classes | | | | |
| Traffic Sign | 1413 | No | 35.82 | 44.54 | 41.88 | 20.40 | 39.54 |
| Light Pole | 9327 | No | 59.40 | 56.49 | 60.61 | 56.10 | 62.13 |
| Fence | 657 | No1 | 30.35 | 21.65 | 29.21 | 22.39 | 25.82 |
| Building | 2910 | No | 29.56 | 30.53 | 31.29 | 25.39 | 27.32 |
| Billboard | 5023 | No | 53.10 | 58.08 | 52.94 | 51.05 | 53.51 |
| Pedestrian | 1632 | No | 36.49 | 49.35 | 41.77 | 34.53 | 39.92 |
| Car | 5496 | No | 72.84 | 78.10 | 79.31 | 69.97 | 77.60 |
| Bus[2] | 136 | No | 16.28 | 4.74 | 0 | 0.42 | 8.77 |
| Motorcycle | 234 | No | 51.65 | 67.85 | 48.70 | 48.53 | 46.83 |
| Heavy Vehicle | 1347 | No | 46.01 | 55.18 | 50.36 | 27.04 | 41.05 |
| Public Transportation | 2525 | No | 59.06 | 67.66 | 65.55 | 48.78 | 57.16 |
| Tree | 5628 | No | 37.01 | 35.76 | 37.23 | 30.53 | 38.70 |

For qualitative comparison, Table 5 shows output samples from the different object detector variants, summarized as follows:

- The model that does not use data augmentation (model 4) is less accurate than the other models and produce more redundant predictions.
- The model that does not use orientation (model 1), the predicted boxes are bigger than object areas, especially for the "curb" and "marking" classes which are oriented by nature.
- The model that predicts θ directly (model 2), the predicated orientations are not accurate or wrong, as in sample numbers 2, 3, 5, 6, 8, and 9. This model produces lots of redundant object predictions for the same object, but with different orientations. Due to non-accurate orientation predictions, IoU between duplicate object predictions becomes small, especially for objects with narrow width like lane markings, and hence non-maximum suppression cannot detect them as duplicates.
- The model that predicts sin(θ) and cos(θ) (model 3) results into better predictions, the number of duplicate predictions decreases.
- The model that predicts sin(2θ) and cos(2θ) (model 5) results in less redundant predictions due to its accurate orientation angle predictions capacity as in sample numbers 3, 6, 8, and 9.

The above findings combined verifies the research hypothesis, discussed in Section 3.2 that solving the orientation encoding discontinuity problem leads to more accurate orientation detection of undirected objects as clearly shown in sample numbers 3, 5, 6, 8, and 9 in Table 5 of model 5 versus model numbers 2 and 3.

### 5.2. Roadway Safety Features Detection

Table 6 reports the roadway safety features extraction accuracy by comparing the system final predictions on the test data with the recorded ground-truth. Overall, the average accuracy for detection is 84.39%, and it ranges between 91.12% and 63.12%. The most accurate feature detected is the 'light pole presence' and the worst is the 'heavy vehicle presence'. This strongly correlates with the feature corresponding objects' appearance in the dataset; the 'light pole' object appeared 9237 times in our dataset, while the 'heavy vehicle' object appeared only 1347 times. QGIS has been used to visualize resulted KSI's of each roadway safety feature as shown in Figure 6.

**Table 5.** Object detection results on random samples from the test data (samples unseen during training time).

| Sample ID | Ground-Truth Objects | Predicted Objects of Different Models | | | | |
|---|---|---|---|---|---|---|
| | | Model 1 (No Orientation) | Model 2 ($\theta$) | Model 3 ($\sin(\theta)$ and $\cos(\theta)$) | Model 4 ($\sin(2\theta)$ and $\cos(2\theta)$ without Augmentation) | Model 5 ($\sin(2\theta)$ and $\cos(2\theta)$) |
| 1 | | | | | | |
| 2 | | | | | | |
| 3 | | | | | | |
| 4 | | | | | | |
| 5 | | | | | | |
| 6 | | | | | | |
| 7 | | | | | | |
| 8 | | | | | | |
| 9 | | | | | | |

Figure 6 shows a topological map visualizing the automatic safety scoring predictions for each roadway safety feature. In Figure 6a, the traffic condition on all primary highways, arterial roads, and collector roads is predicted to be fluid, because the vast majority of test data was captured over weekends. The majority of the test data was captured across rural areas except most of the Ring Road which runs through urban areas in the most part. The system predictions confirm this fact, as shown in Figure 6b. Most of the travelled roads in the test dataset have a median, the high accuracy of road separation feature detection is indicated by the green color (high safety factor) in the vast majority of roads in Figure 6c. Predictions visualized in Figure 6d,e confirm the lack of road markings and traffic signs in most of the covered roads; which indicates a needed intervention to enhance roadway safety. Roadside barriers need to be maintained along the Ring Road and need to be constructed along arterial and collector roads, as shown in Figure 6f. As shown in

Figure 6g, all main roads in GCR are witnessing multitude of billboards which can be a source of driver distraction. As shown in Figure 6h, the system detects an alarming number of pedestrian waiting areas along highways and expressway (particularly the Ring Road), in fact, most of them are not official and there is a need to study their impact on crash numbers. As shown in Figure 6i, the system identifies a large number of heavy vehicles along a regional primary highway, which negatively attributes to roadway safety. As shown in Figure 6j, all roads inside GCR are detected to be covered with light poles which aligns with the urban peripheries of the study area. The resulted maps demonstrate the effectiveness of the proposed system by identifying the roadway segments that have the potential of high safety gains (or issues thereof) to better inform decision makers focusing their efforts on these much-needed features that requires specific interventions.

**Table 6.** Roadway safety features estimation accuracy.

| Roadway Safety Feature | Prediction Accuracy |
|---|---|
| Traffic | 81.70% |
| Area Type | 85.87% |
| Road Separation | 89.91% |
| Marking Presence | 90.85% |
| Traffic Signs Presence | 86.14% |
| Roadside Barrier Presence | 82.64% |
| Billboard Presence | 82.23% |
| Pedestrian Waiting Area Presence | 90.31% |
| Heavy Vehicle Presence | 63.12% |
| Light Pole Presence | 91.12% |

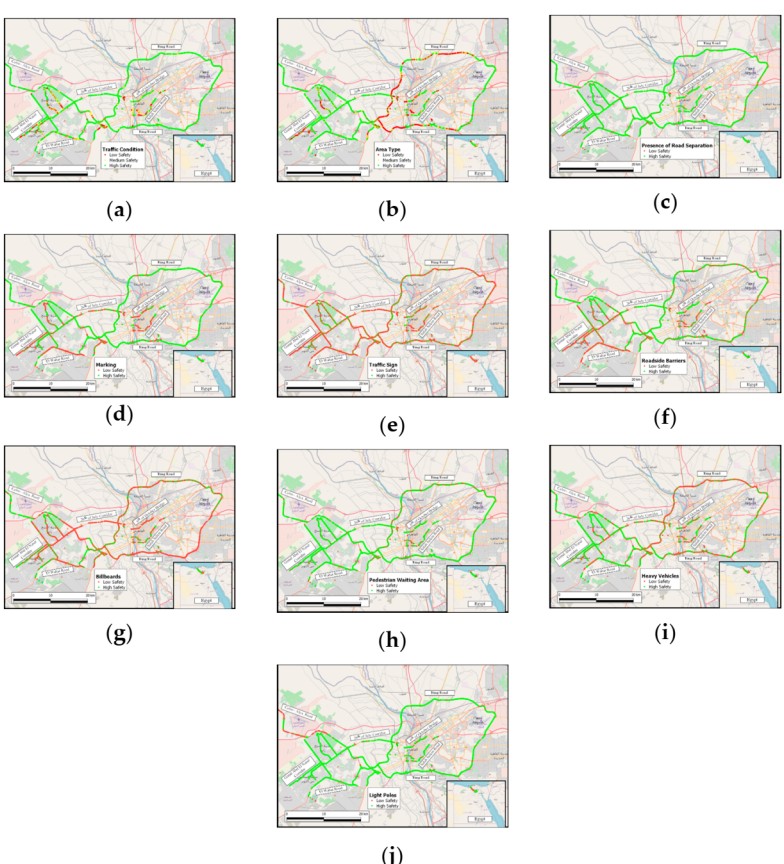

**Figure 6.** Automatically detected roadway features maps for: (**a**) traffic condition, (**b**) area types, (**c**) road separation, and presence of (**d**) marking, (**e**) traffic signs, (**f**) roadside barrier, (**g**) billboard, (**h**) pedestrian waiting area, (**i**) heavy vehicles, (**j**) and light poles.

## 6. Conclusions and Future Work

Different roadway safety features may have a direct impact on safety. Extensive research has been conducted towards quantifying this impact. The most essential stage in the process of roadway safety management is data collection that make the process expensive and time-consuming, especially in developing and low-income countries, where the issues of roadway safety is exacerbated. Therefore, the automation of roadway safety features detection process is a promising and much-needed application. In this research, a new pervasive system for automatic road safety features detection is proposed to make the process more applicable and less costly. Given a front-facing camera and a GPS sensor, the system automatically evaluates ten roadway safety features. A modification to the state-of-the-art deep learning-based oriented object detection model is introduced, which improved its accuracy by 16.9% of mAP. Additionally, different oriented object detection strategies are presented and analyzed. The system includes a rule-based module that processes the object detection output to provide ten roadway safety features. The rule-based module does not guarantee that an absent road feature causes the estimated road segment safety score to be very low and is geographically dependent, further research work can be focused upon improving it.

The system can be used to pervasively enable or disable autonomous driving (AD) or advanced driver assistance systems (ADAS) and could aspire for "safety-based navigation" applications where travelers can choose not only faster and shorter routes, but also safer routes. To train and validate the proposed approach, a dataset for roadway safety features extraction in developing countries is introduced: Roadway Safety Dataset. The dataset is composed of 473 km of driving, covering local roads, primary collector roads, primary and secondary arterial roads, and regional and primary freeways. The proposed approach is extensively evaluated against the dataset, and encouraging results are achieved that is presented as a benchmark. The average roadway safety feature prediction accuracy is 84.39% and ranges between 91.11% and 63.12%. Finally, topological maps over GCR showing the proposed system automatic detection of each roadway safety feature are presented and analyzed. The resulted maps demonstrate the effectiveness of the proposed approach by showing areas that have safety issues to make decision makers focus on the features that need more development.

While this research attempted to answer specific questions, it triggered even questions with abundant room for extensions and future work. Firstly, our solution can be used in ranking road safety according to the detected safety-related features using multi-criteria decision-making methods in case of lack of crash and safety data. Secondly, applying camera calibration can allow more accurate estimation of geometric dimensions of the detected road objects. Additionally, the system could aspire for "safety-based navigation" applications where travelers can choose not only faster and shorter routes, rather safer routes.

**Author Contributions:** Conceptualization, H.A., H.M.E. and D.S.; methodology, H.M.E. and M.N.M.; software, H.M.E., K.S. and O.R.E.; validation, H.M.E., K.S. and O.R.E.; investigation, H.A., H.M.E. and D.S.; resources, H.A. and H.M.E.; data curation, K.S.; writing—original draft preparation, H.M.E. and K.S.; writing—review and editing, H.A. and D.S.; visualization, H.M.E. and K.S.; supervision, H.A., D.S. and M.N.M.; project administration, H.A. and D.S.; funding acquisition, H.A. and H.M.E. All authors have read and agreed to the published version of the manuscript.

**Funding:** This research received no external funding.

**Institutional Review Board Statement:** Not applicable.

**Informed Consent Statement:** Not applicable.

**Conflicts of Interest:** The authors declare no conflict of interest.

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
