# Peer review of "Automatic Roadway Features Detection with Oriented Object Detection"

_applsci, doi:10.3390/app11083531_

Round 1

Reviewer 1 Report

The work is very solid and well presented. The presented approach shows good results in object detection using a deep-learning AI. It provides a good contribution for future scientific works related to road safety. 

The authors, however mention "safety fusion", where vehicle could eventually share their knowledge about the road objects they detect. This "safety fusion" approach can work if there is a validated entity, such as the motorway infrastructure operator, for example, that can receive and validate the vehicle information, otherwise security might be compromised. 

The authors also refer that their work could contribute for future marking of road blackspots. I suggest that this should be better explained.

In overall, the authors provided a work that deserve merit.

Author Response

Reply to Reviewers’ Suggestions

March 29th, 2021

The authors kindly appreciate the reviewers’ thorough and insightful feedback. We have done our utmost best to address all the comments and suggestions and we feel our article is now stronger.

Reply to Reviewer 1:

Thank you for your insightful comments and suggestions that are helpful to improve the manuscript. The followings are our responses to each of your comments.

  1. The work is very solid and well presented. The presented approach shows good results in object detection using a deep-learning AI. It provides a good contribution for future scientific works related to road safety.
    The authors, however mention "safety fusion", where vehicle could eventually share their knowledge about the road objects they detect. This "safety fusion" approach can work if there is a validated entity, such as the motorway infrastructure operator, for example, that can receive and validate the vehicle information, otherwise security might be compromised.

Our Answer: Thank you for pointing this insightful note out. We added the following to the paper in the paragraph discussing figure 2: “To reduce information security risks, a validation entity such as a motorway infrastructure operator can receive and validate the vehicles’ shared information.”.

  1. The authors also refer that their work could contribute for future marking of road blackspots. I suggest that this should be better explained.
    In overall, the authors provided a work that deserve merit.

Our Answer: Thank you for your suggestions and that comment. The paper is updated to define the black spots identification to be based on when the detected safety features scores are lower than the roadway safety standards.

Reviewer 2 Report

The study is very important particularly for developing communities where road accidents or crashes are rampant, which I believe with assistive technologies embedded in vehicles can potentially minimise accidents occurrence. However, the authors should address the following questions/observations

  1. What is the proposed novelty in this article?
  2. In what way(s) this research outperformed the previous studies. Detail comparison should be provided.
  3. Must all the mentioned features in this article have to be present or absent for a road to be considered safe? Authors are expected to elaborate on this.
  4. Finally, the article requires thorough proofreading to meet the standard of this journal.
  5. "... A deep Learning vision-based" in the title, I suggest you find an appropriate statement to replace that.

Author Response

Reply to Reviewer 2 Suggestions

March 29th, 2021

The authors kindly appreciate the reviewers’ thorough and insightful feedback. We have done our utmost best to address all the comments and suggestions and we feel our article is now stronger.

Reply to Reviewer 2:

Thank you for your insightful comments and suggestions that are helpful to improve the manuscript. The followings are our responses to each of your comments.

The study is very important particularly for developing communities where road accidents or crashes are rampant, which I believe with assistive technologies embedded in vehicles can potentially minimise accidents occurrence. However, the authors should address the following questions/observations

  1. What is the proposed novelty in this article?
    Our Answer: Thank you for your comment. The main proposed novelty of this article is our introduced  pervasive system, and approach, that automates the process of road safety estimation. Given a front-facing camera and a GPS, which exist in most of modern cars, roadway safety scores are estimated automatically based on diverse roadway features. The system utilizes a new oriented (rotated) object detection model that solves an 'orientation encoding discontinuity' problem to improve detection mean Average Precision (mAP) by 16.9% compared to object detection methods in literature.
    The system can be used to pervasively enable or disable Autonomous Driving (AD) or Advanced Driver Assistance Systems (ADAS) and could aspire for "safety-based navigation" applications where travelers can choose not only faster and shorter routes, but also safer routes. 
    A fully-annotated dataset for roadway safety features extraction was collected covering 473 km of roads.
    We improved the conclusion to be more focused in highlighting the proposed novelty.

  2. In what way(s) this research outperformed the previous studies. Detail comparison should be provided.
    Our Answer: Thank you for pointing this out. It is important to highlight the ways this research outperformed the previous studies. We added references to the results comparing our object detection model results with previous models to Tables 3, 4, and 5.
  3. Must all the mentioned features in this article have to be present or absent for a road to be considered safe? Authors are expected to elaborate on this.Our Answer: Thank you for your insightful comments. We use a rule-based method of weighted averaging the safety scores over the different road features in a road segment to estimate its safety score. The score is a contentious number normalized to be from 0 to 1 (for a very safe segment). An absent road feature doesn't guarantee that the estimated road segment safety score to be very low, and hence such rule-based method can be improved.
    We improved the description in section 3.5 and highlighted the potential improvement in the future work section: "The rule-based module does not guarantee that an absent road feature causes the estimated road segment safety score to be very low, and hence such rule-based method can be improved.".

  4. Finally, the article requires thorough proofreading to meet the standard of this journal.
    Our Answer: Thank you for the useful feedback. We made a thorough language check and made corrections and improvements to the writing and the tables presentation.

  5. "... A deep Learning vision-based" in the title, I suggest you find an appropriate statement to replace that.
    Our Answer: Thank you for the useful suggestion. We changed the title to be clearer to: "Automatic Roadway Features Detection with Oriented Object Detection".

Reviewer 3 Report

An oriented bounding box object detection model, introduced in the paper, seems to be new in the existing technique in the field of Systems, based on Deep Learning and vision-based data. I am convinced that this idea is an important contribution to the development of artificial intelligence tools designed to support the safety of autonomous vehicles. The validation of the efficiency of the improved algorithm is convincing too.The state of development of this field of knowledge and the techniques used are well presented. 

The weakest point of the whole algorithm seems to be the rule-based module. It is obvious that this system is geographically dependent. Usually, however, such systems are biased due to the different ways of building rules and the bias of experts. This module is not the subject of the article, but a deeper discussion of its effectiveness and impact on the effectiveness of the entire system would be interesting

Author Response

Reply to Reviewer 2 Suggestions

March 29th, 2021

The authors kindly appreciate the reviewers’ thorough and insightful feedback. We have done our utmost best to address all the comments and suggestions and we feel our article is now stronger.

Reply to Reviewer 3:

Thank you for your insightful comments and suggestions that are helpful to improve the manuscript. The followings are our responses to each of your comments.

An oriented bounding box object detection model, introduced in the paper, seems to be new in the existing technique in the field of Systems, based on Deep Learning and vision-based data. I am convinced that this idea is an important contribution to the development of artificial intelligence tools designed to support the safety of autonomous vehicles. The validation of the efficiency of the improved algorithm is convincing too. The state of development of this field of knowledge and the techniques used are well presented. 

The weakest point of the whole algorithm seems to be the rule-based module. It is obvious that this system is geographically dependent. Usually, however, such systems are biased due to the different ways of building rules and the bias of experts. This module is not the subject of the article, but a deeper discussion of its effectiveness and impact on the effectiveness of the entire system would be interesting.

Our Answer: Thank you for your comments and for your insightful note regarding the adopted rule-based module. We improved the method description in section 3.5. Given that this rule-based module is not the subject of the article, we highlighted such an important improvement in the Future Work section:

"The rule-based module does not guarantee that an absent road feature causes the estimated road segment safety score to be very low and is geographically dependent, further research work can be focused upon improving it".